# Estimating associations between antidepressant use and incident mild cognitive impairment in older adults with depression

Fang Han[1,2,3]☯*, Tyler Bonnett[4]☯, Willa D. Brenowitz[5], Merilee A. Teylan[2], Lilah M. Besser[2], Yen-Chi Chen[2], Gary Chan[2], Ke-Gang Cao[3], Ying Gao[3], Xiao-Hua Zhou[6]*

**1** Department of Acupuncture and Moxibustion, Beijing Hospital of Traditional Chinese Medicine, Capital Medical University, Beijing, China, **2** National Alzheimer's Coordinating Center, Department of Epidemiology, University of Washington, Seattle, Washington, United States of America, **3** Department of Neurology, Dongzhimen Hospital Affiliated to Beijing University of Chinese Medicine, Beijing, China, **4** Clinical Monitoring Research Program Directorate, Frederick National Laboratory for Cancer Research sponsored by the National Cancer Institute, Frederick, Maryland, United States of America, **5** Department of Psychiatry, University of California, San Francisco, California, United States of America, **6** Department of Biostatistics, Beijing International Center for Mathematical Research, Peking University, Beijing, China

☯ These authors contributed equally to this work.
* 1989hanfang@sina.com (HF); azhou@math.pku.edu.cn (Z-XH)

## Abstract

### Introduction

Previous studies have provided equivocal evidence of antidepressant use on subsequent cognitive impairment; this could be due to inconsistent modeling approaches. Our goals are methodological and clinical. We evaluate the impact of statistical modeling approaches on the associations between antidepressant use and risk of Mild Cognitive Impairment (MCI) in older adults with depression.

### Methods

716 participants were enrolled. Our primary analysis employed a time-dependent Cox proportional hazards model. We also implemented two fixed-covariate proportional hazards models—one based on having <u>ever</u> used antidepressants during follow-up, and the other restricted to <u>baseline</u> use only.

### Results

Treating antidepressant use as a time-varying covariate, we found no significant association with incident MCI (HR = 0.92, 95% CI: 0.70, 1.20). In contrast, when antidepressant use was treated as a fixed covariate, we observed a significant association between having <u>ever</u> used antidepressants and lower risk of MCI (HR = 0.40, 95% CI: 0.28, 0.56). However, in the <u>baseline-use</u> only model, the association was non-significant (HR = 0.84, 95% CI: 0.60, 1.17).

**Data Availability Statement:** The data used in the manuscript can be applied on the website of https://www.alz.washington.edu/.

**Funding:** The NACC funding (U01 AG016976) from NIH/NIA supported overall data collection. This

project has been funded in whole or in part with federal funds from the National Cancer Institute, National Institutes of Health, under Contract No. HHSN261200800001E. The funders had no role in study design, data collection and analysis, decision to publish, or preparation of the manuscript.

**Competing interests:** The authors have declared that no competing interests exist.

## Discussion

Our results were dependent upon statistical models and suggest that antidepressant use should be modeled as a time-varying covariate. Using a robust time-dependent analysis, antidepressant use was not significantly associated with incident MCI among cognitively normal persons with depression.

## Introduction

Depression and dementia are both common in older adults[1–5]. Previous studies have suggested that depression may be a risk factor for dementia[6] or may be prodromal to dementia [7–8]. Findings about associations between antidepressant use and subsequent cognitive impairment have been inconsistent. Several studies have suggested that antidepressant use is associated with increased risk of developing cognitive impairment and dementia[9–14]. Some studies have found an inverse relationship—that antidepressant use may decrease the risk of developing dementia[15–16]. In some cases researchers have differentiated among classes of antidepressants, such as tricyclic antidepressants (TCAs), monoamine oxidase inhibitors (MAOIs), selective serotonin reuptake inhibitors (SSRIs), and serotonin and noradrenaline reuptake inhibitors (SNRIs)[17–24]. Conclusive evidence on whether antidepressant use influences cognitive function is still lacking.

Methodological approaches to estimating the association between antidepressant use and cognitive decline vary in these prior studies, as do definitions of "exposure" (i.e., antidepressant use) and "outcome" (i.e., cognition). We hypothesize that the multitude of statistical procedures, the differing criteria used to define depression and the wide variety of approaches to characterizing antidepressant use by previous studies may contribute to the inconsistent estimates of the association that have obtained. Furthermore, we suggest that implementation of time-fixed models which are sensitive to immortal time bias could be a major limitation of previous work. Despite the well-documented shortcomings of these models, their use persists [25].

We have two goals of this study one clinical goal: to evaluate the association between antidepressant use and cognitive impairment and one methodological goal: to illuminate the benefits and pitfalls of different methodological approaches and apply relatively robust methods and definitions for inference. We provide a direct comparison of the results of a time-fixed model and a time-dependent model, the latter of which we suggest is the appropriate method for handling this data, acknowledging that our results would inevitably be in conflict with some others.

## Methods

### 2.1 Data source

Data used in this study comes from the National Alzheimer's Coordinating Center (NACC), which maintains a database representing the clinical enrollment of the 39 past and present Alzheimer's Disease Centers (ADCs) supported by the U.S. National Institute on Aging/National Institutes of Health. In those centers, participants underwent annual evaluations according to a standardized protocol, the Uniform Data Set (UDS), described in detail elsewhere[26–27]. Many of the subjects represented in the NACC database were volunteers referred by themselves, family, or friends due to concerns about their memory. Written informed consents

were obtained from all participants at individual ADCs where they were enrolled. Research using the NACC database was approved by the University of Washington Institutional Review Board.

## 2.2 Study sample

Participants were eligible for this study if they were aged 60 years or older, cognitively normal, and depressed at their first UDS visit. Participants also must have made at least three UDS visits from 2005 to 2016. Assessment of normal cognition and MCI was made by either a single clinician or a formal consensus panel at each ADC. Subjects were considered depressed if they met at least two of the following five criteria at entry into the study: 1) self-reported active depression in the last two years, 2) depression or dysphoria symptoms as reported by a co-participant on the Neuropsychiatric Inventory Questionnaire (NPI-Q)[28], 3) Geriatric Depression Scale-15 (GDS-15) score of at least six[29,30], 4) clinically depressed mood based on clinician interview, or 5) a clinical diagnosis of active depression based on current UDS examination and the clinician's best judgment.

Fig 1 shows that there were 11,096 participants aged 60 years or older and cognitively normal at baseline. Among those, 6,184 made at least three UDS visits. Finally, 716 participants met two of the five above depression criteria. The majority of participants met depression criteria by self-reported active depression in the last two years (90.6%) in addition to at least one other criteria; a full breakdown of the number of participants who met each respective criterion is presented in S1 Table.

All participants included in this study received UDS clinical evaluations at baseline and yearly thereafter.

## 2.3 Primary outcome and study overview

The primary outcome was first diagnosis of MCI. MCI represents a definable pre-dementia stage of the continuum of cognitive decline. Some persons diagnosed with MCI will not progress to dementia; however, the use of incident MCI as outcome may also yield a higher event rate and more study power within the time period. Diagnosis of MCI was made according to the Petersen criteria if the subject did not have normal cognition and was not clinically demented, but had cognitive complaints not normal for their age, and had largely preserved independence in functional activities[31,32]. MCI is a transitional state characterizing cognitive decline; therefore, if a participant was diagnosed with dementia directly from normal cognition without an intervening MCI diagnosis, then an intermediate MCI stage was assumed to have occurred at the midpoint between the participant's last normal cognition visit and the visit at which a diagnosis of dementia was made. Such participants were included in the analyses.

## 2.4 Assessment of antidepressant exposure

Use of medication, including antidepressants, was captured as part of the UDS clinical evaluation, based on clinical interview. Participants were asked to report all prescription medications taken within the two weeks before the current visit.

## 2.5 Additional clinical measurements

Information on a variety of relevant covariates was collected by the UDS. Demographic characteristics including age, sex, race, and years of education were documented at entry into the study. UDS participants provide a detailed health history (UDS Form A5), completed by a

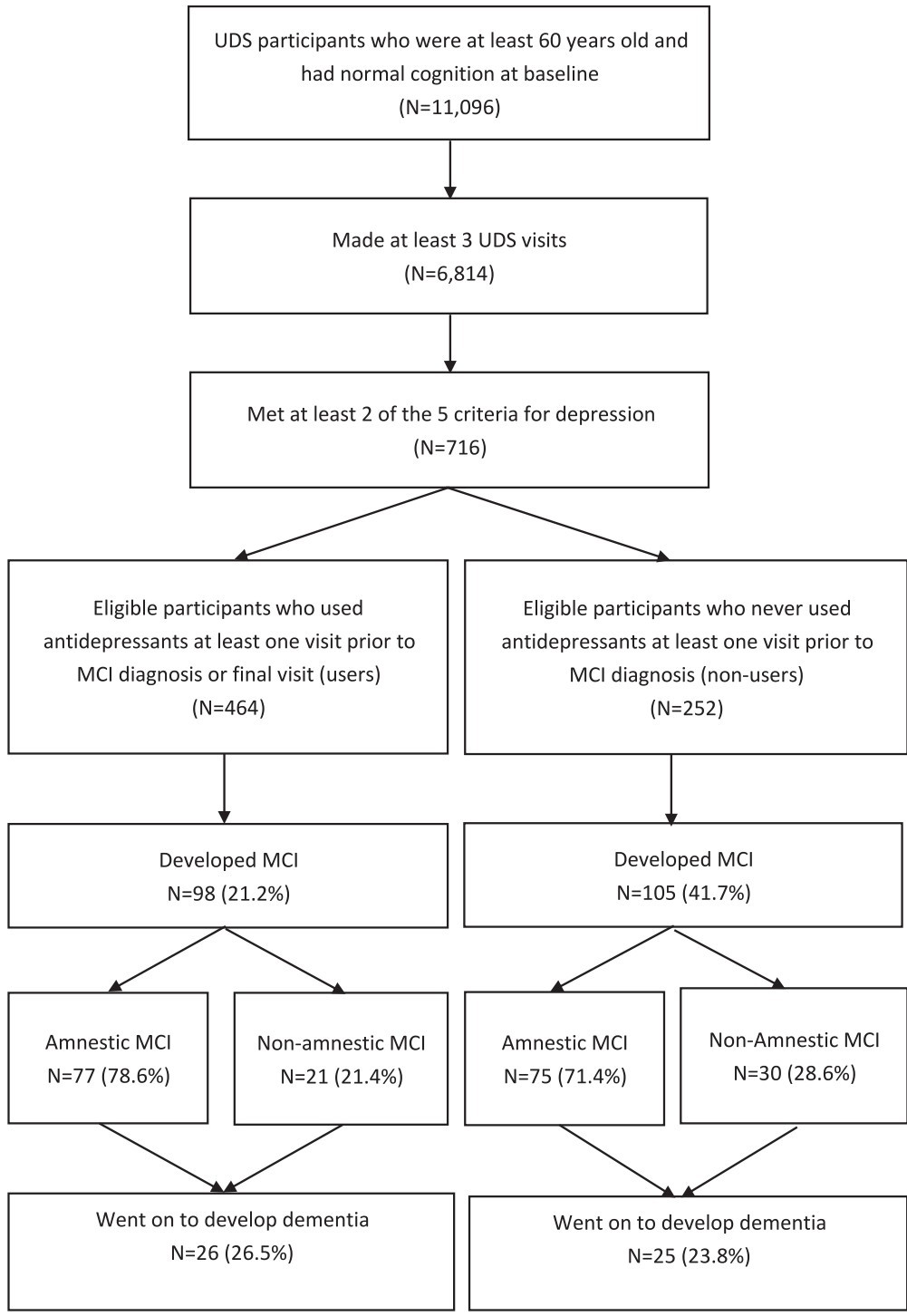

**Fig 1. Flowchart of study participant selection process.**

clinician based on the subject's report, medical records, and observation using the clinician's best judgement. We defined a list of comorbidities that may confound the relationship among depression, antidepressant use, and cognitive decline, and adjusted in our analyses for history of diabetes, hypertension, hypercholesterolemia, and cardiovascular disease. We adjusted for

cigarette smoking based on an initial-visit report of having smoked more than 100 cigarettes in the participant's lifetime. We also adjusted for apolipoprotein E (APOE) genotype in our models by indicating the presence or absence of at least one e4 allele, a known risk factor for the development of MCI and dementia[33–36].

## 2.6 Statistical analysis

To investigate the association between antidepressant use and incident MCI, we implemented two types of proportional hazards regression models. The first was a time-dependent Cox proportional hazard model in which antidepressant use was allowed to change over time according to participants' responses at follow-up visits. This time-dependent approach avoids the potential for immortal time bias. The other type of model used was a fixed-covariate model, in which antidepressant use was categorized and did not change during follow-up. Participants were categorized as either users or non-users of antidepressants in two separate fixed-covariate models: (a) use or non-use of antidepressants at any visit prior to MCI diagnosis visit; (b) use/non-use of antidepressants at baseline, i.e., first visit.

Fixed-covariate model (a) has a major drawback in its susceptibility to immortal time bias: i.e., participants with longer follow-up times will be more likely to eventually use antidepressants and thus be categorized as users. By contrast, fixed-covariate model (b) oversimplifies participants' real patterns of antidepressant use over time but avoids the possible complications posed by immortal time bias. Immortal time bias is common in studies where participants can encounter the exposure during follow-up. This bias can artificially inflate estimation of the survival time of participants in the exposure group, and therefore could bias results to make antidepressant use appear to reduce the hazard of incident MCI in the absence of a true association. In this model, participants would be categorized as antidepressant users only if the use was reported before the visit where the MCI diagnosis was made. This practice is designed to avoid including participants who began taking antidepressants because of early symptoms associated with MCI.

Finally, we conducted sensitivity analyses to examine the following: (1) How would changing our definition of depression, e.g., from meeting at least two criteria to meeting at least three criteria, affect results? (2) How would simply defining depression as a GDS score of 6 or greater affect the results?

All models described in this study adjusted for age, sex, race, level of education, comorbidity history, smoking history, and the presence of the e4 allele in the APOE genotype. All analyses were conducted using the statistical programming software R (version 3.3.2). For each model, we computed hazards ratios (HRs) and 95% confidence intervals (95% CIs) to assess the risk of developing MCI. The proportional hazards assumption was verified using the Schoenfeld Test [37].

## Results

In total, 716 participants met the inclusion criteria for our study (Fig 1). Of those, 464 (64.8%) reported using antidepressants at least once at a visit prior to MCI diagnosis or their final visit (ever-users), and 252 (35.2%) never reported antidepressant use (never-users). Participants had an average length of follow-up of five years in this study. Antidepressant ever-users were slightly younger on average, more likely to be female, less likely to identify as non-white, more likely to have a history of smoking and hypercholesterolemia. Ever-users tended to be consistent users. Of the 464 ever-users, 422 (90.9%) reported antidepressant use during at least half of their UDS visits, and 288 (62.1%) reported using antidepressants at all visits during follow-up. Of the 716 participants, 413 (57.7%) were baseline antidepressant users and 303 (42.3%)

**Table 1. Characteristics of study sample by antidepressant use (N = 716).**

| Characteristic Mean (SD) or n (%) | Antidepressant Use | | | |
|---|---|---|---|---|
| | Ever-Users (n = 464) | Never-Users (n = 252) | Baseline Users (n = 413) | Baseline Non-users (n = 303) |
| Age (years) | 71.9 (7.5) | 73.4 (7.9) | 72.0 (7.4) | 73.1 (8.1) |
| Female | 353 (76.1%) | 155 (61.5%) | 308 (74.6%) | 200 (66.0%) |
| College degree or higher | 279 (60.1%) | 135 (53.6%) | 246 (59.6%) | 168 (55.4%) |
| Non-White Race | 53 (11.4%) | 44 (17.4%) | 41 (9.9%) | 56 (18.5%) |
| 1+ APOE-e4 Alleles | 126 (27.2%) | 61 (24.2%) | 114 (27.6%) | 73 (24.1%) |
| Number of visits | 5.3 (2.1) | 5.2 (2.1) | 5.2 (2.1) | 5.4 (2.1) |
| Duration of follow-up (years) | 5.1 (2.4) | 5.0 (2.3) | 5.0 (2.3) | 5.2 (2.3) |
| Smoker (≥ 100 lifetime cigarettes) | 235 (50.6%) | 108 (42.8%) | 211 (51.1%) | 132 (43.6%) |
| Baseline GDS-15[a] | 3.2 (3.1) | 4.1 (3.3) | 3.0 (3.0) | 4.1 (3.6) |
| Hypertension | 237 (51.1%) | 135 (53.6%) | 215 (52.1%) | 157 (51.8%) |
| Diabetes | 61 (13.1%) | 40 (15.9%) | 52 (12.6%) | 49 (16.2%) |
| Hypercholesterolemia | 250 (53.9%) | 109 (43.3%) | 225 (54.5%) | 134 (44.2%) |
| Cardiovascular Disease | 136 (29.3%) | 78 (31.0%) | 122 (29.5%) | 92 (30.4%) |

[a] Geriatric Depression Scale-15

were baseline non-users. A summary of baseline characteristics for participants based on their patterns of antidepressant use is given in Table 1.

Among the 464 ever-users, 98 (21.2%) eventually developed MCI, compared to 105 of the 252 never-users (41.7%). There were 26 participants (20 ever-users and 6 never-users) who went directly from normal cognition to dementia without an intermediary MCI diagnoses, although for these participants an intermediate MCI stage was assumed to have occurred at the midpoint between their latest normal-cognition visit and their first diagnosis of dementia.

In our primary analysis which utilized a time-dependent Cox proportional hazards model, we did not find an association between antidepressant use and risk of developing MCI (HR = 0.92, 95% CI: 0.70, 1.20; Table 2). However, the association changed when we treated antidepressant use as a fixed covariate. In the first fixed-covariate model, antidepressant ever-users appear to have significantly decreased risk of developing MCI (HR = 0.40; 95% CI: 0.28, 0.56; Table 2). However, when we grouped according to baseline antidepressant use, there was no significant difference between baseline users and non-users (HR = 0.84; 95% CI: 0.61, 1.17; Table 2).

**Table 2. Antidepressant use and risk of developing MCI[*].**

| Model and Model Setting | HR for antidepressant exposure | 95% CI |
|---|---|---|
| **Primary analyses** [a] | | |
| Time-varying covariate model | 0.92 | 0.70, 1.20 |
| Fixed-covariate model (ever-use vs. never-use) | 0.40 | 0.28, 0.56 |
| Fixed-covariate model (baseline use vs. baseline non-use) | 0.84 | 0.61, 1.17 |

[*] Adjusted for age, sex, race, level of education, comorbidity history (diabetes, hypertension, hypercholesterolemia, and cardiovascular disease), smoking history, and the presence of the APOE e4 allele.

[a] In these primary models, antidepressant use was required to have occurred at least one visit prior to MCI diagnosis or final UDS visit. Participants were considered depressed at entry if they met two of the five criteria for depression.

**Table 3. Antidepressant use and risk of developing MCI—Sensitivity analyses*.**

| Model and Model Setting | HR for antidepressant exposure | 95% CI |
|---|---|---|
| **3+ Depression Criteria Required for Entry** | | |
| Time-varying covariate model | 0.69 | 0.47, 1.03 |
| Fixed-covariate model (ever-use vs. never-use) | 0.23 | 0.13, 0.38 |
| Fixed-covariate model (baseline use vs. baseline non-use) | 0.67 | 0.41, 1.10 |
| **GDS $\geq$ 6 Required for Entry** | | |
| Time-varying covariate model | 1.18 | 0.75, 1.86 |
| Fixed-covariate model (ever-use vs. never-use) | 0.49 | 0.26, 0.92 |
| Fixed-covariate model (baseline use vs. baseline non-use) | 0.99 | 0.54, 1.80 |

*Adjusted for age, sex, race, level of education, comorbidity history (diabetes, hypertension, hypercholesterolemia, and cardiovascular disease), smoking history, and the presence of the APOE e4 allele.

Finally, we performed two sensitivity analyses to address the impact of depression definition. Inference in these sensitivity analyses were the same as above, however point estimates changed. First, we required that subjects meet at least three depression criteria rather than two. This resulted in a population of 307 subjects who met the revised criteria for depression. A breakdown of the number of participants who met each of the criteria, out of all those who met at least three, is presented in S2 Table. The hazard ratio comparing users to non-users shrank from 0.92 in the original model to 0.69 in this sensitivity analysis (95% CI: 0.47, 1.03; Table 3). In the fixed-covariate model comparing ever-users and never-users, we observed a similar reduction in the estimated hazard ratio from 0.40 in the primary analysis to 0.23 (95% CI: 0.13, 0.38; Table 3). In the fixed-covariate model comparing baseline users and baseline non-users, the estimated hazard ratio shrank from 0.84 in the primary setting to 0.67 in this analysis (95% CI: 0.41, 1.10; Table 3). We then ran an analysis where the definition of depression was based solely on the participants' baseline GDS score being at least 6 (which reduced the population of analysis-eligible participants to 231). We observed slightly larger hazard ratios in the time-dependent model (HR = 1.18; 95% CI: 0.75, 1.86; Table 3), the fixed-covariate model comparing ever-users to never-users (HR = 0.49; 95% CI: 0.26–0.92; Table 3), and the fixed-covariate model comparing baseline users to baseline non-users (HR = 0.99; 95% CI: 0.54, 1.80; Table 3).

## Discussion

We did not find a significant association between antidepressant use and risk of incident MCI in our primary analysis which utilized the time-dependent Cox proportional hazards model. We consider this the primary clinical outcome of our work. We also directed our work toward a methodological goal. Previous work has shown that the time-dependent model is the appropriate approach to hazard-based analyses when the exposure of interest can occur during follow-up, most notably because it avoids the potential for immortal time bias that is inherent in these settings[38]. The goal of our implementation of time-fixed models was to provide a practical example of the direction and magnitude of differences which can be obtained based on a modelling choice that invites bias and yet persists in hazards-based analyses [25]. We found that the bias of time-fixed covariate models can be large and varies widely depending on how the time-fixed covariate is defined. When we defined antidepressant use based on ever having

reported use, the time-fixed model estimated a significant protective effect on cognitive decline: a hazard ratio of 0.40 as opposed to the hazard ratio of 0.92 estimated by the time-dependent model. Were we to believe the smaller hazard ratio, the overestimation of the effect would enough to misdirect future research. If we define antidepressant use via baseline use we tend to estimate a hazard ratio closer to that of the time-dependent model (0.84 versus 0.92 in our primary comparisons), but the more simplistic model still tends to overestimate the effect. We saw this pattern repeated in sensitivity analyses where the threshold for depression was altered. We hope that this provides an illustration of the impact that naïve modelling choices can have. Future researchers should insist on using time-dependent models to study this association.

We also observed, perhaps predictably, that changing the threshold for depression did impact the results of our analyses. The goal of these sensitivity analyses was to highlight the magnitude of the differences which can be obtained. The comprehensive nature of the UDS allowed us to make these considerations. If we required participants to meet at least three (rather than two) the components of our definition, estimated hazard ratios provided by the three models decreased by 29.2% on average. On the other hand, if we only required that each participant had a baseline GDS score of at least six, hazard ratios increased on average by 22.9%. Even though the statistical significance of the hazard ratios was unchanged in all three models under both sensitivity analysis settings, future studies should be careful to evaluate the severity of depression and be aware of the non-negligible effect that including severely depressed participants or patients with mild depression can have.

We have addressed strategies for researchers who choose to perform (or as a result of available data are driven to) hazards-based analyses. Several prior studies have used hazard-based analyses to examine the association between anti-depressant use and cognitive impairment or dementia[10–12,14,16]. Results are inconclusive; several studies found that antidepressant use is associated with increased risk of developing cognitive impairment and dementia[10–12,14], while another found that use was associated with a decreased risk of developing dementia[16]. None of these studies used time-dependent analyses. Our study extends upon these findings, suggesting no association between anti-depressant use and incident MCI when using robust time-dependent hazard analyses. Goveas et al. conducted a similar analysis in the Women's Health Initiative Memory Study (WHIMS) but found that antidepressant use was associated with increased risk of MCI [12]. In this study, antidepressant use was assessed according to their current medications and participants were grouped into antidepressant users/non-users at baseline. This is in contrast to our findings when only use anti-depressant at baseline, which may be due to differing study populations (only healthy postmenopausal women were included in WHIMS) or exposure or outcome definitions. Several studies also summarized the relationship between antidepressant use and cognitive decline using longitudinal rates of change (as is in the case in linear mixed model approaches) [19,20] or odds ratios (provided by logistic regression) [17]. Recent studies that have used the linear mixed modeling approach failed to find evidence for a significant association between antidepressant use and cognitive decline[19–21]. In these models, study participants are assigned a cognitive score (usually based on a battery of cognitive tests) and the primary comparison is made between antidepressant users and non-users with respect to the rate of change on this score. These results agree with our finding that the risk of incident MCI faced by antidepressant users and non-users is not significantly different. However, it should be noted that cognitive test batteries may not be sensitive enough to detect true differences in rates of decline between users and non-users, even if they did exist.

On the other hand, it is more difficult to compare our results with those using logistic regression. A previous meta-analysis found that antidepressant users had increased odds of

Alzheimer's disease relative to non-users (OR = 1.65) [9]. Importantly, our work investigated MCI as the outcome of interest and not Alzheimer's disease. The difference between the conclusion reached here and in the Moraros et al. meta-analysis is large; however, it might be that the cognitive outcome assessed also plays a key role in studies of this association. If that is the case, then it may not necessarily be a contradiction to report that antidepressant use has apparently little impact on risk of incident MCI but plays a larger role in development of dementia or Alzheimer's disease. Our results can provide some exploratory evidence toward this conclusion. We found that ever-users of antidepressants were less likely to develop MCI than never-users (21.2% of ever-users developed MCI compared to 41.7% of never-users, without adjustment for potential confounders), and we found similar proportions of both groups went on to receive a dementia diagnosis (26.5% of ever-users and 23.8% of never-users). That is, non-users experienced a similar rate of dementia diagnosis despite experiencing twice the rate of MCI diagnoses. This could be an interesting avenue for future research.

Research is also needed in more diverse populations. The UDS is not a nationally representative sample. Most participants in our study were white and more than half had some college education. Since many subjects in the NACC database were volunteers referred by themselves, friends, or family members due to concerns about their memory, observed rates of MCI in this sample may be higher than in the general. Prior studies have also suggested that late-life depression might be accompanied by cognitive decline[5], hence there is a possibility that some of the more depressed participants may already had very mild cognitive impairment at entry into our study despite being categorized as cognitively normal. In the future, participants from a wider array of backgrounds could be evaluated over longer trajectories to address generalizability of our results.

In conclusion, this study illustrates the potential bias in time-fixed models compared to robust methods that account for time-varying exposures. We did not find an association between antidepressant use and risk of incident MCI in our primary analysis which utilized the time-dependent Cox proportional hazards model. The bias of time-fixed covariate models can be large: when we defined anti-depressant use based on ever having reported use, the time-fixed model estimated a significant protective effect of antidepressant use on cognitive decline. However future research is implemented, researchers should be careful to avoid common modelling mistakes and should treat antidepressant use as a time-varying covariate when the data allows. If data are not longitudinal and do not permit such an approach, researchers should carefully consider this limitation. As a general strategy, researchers should recognize the potential for bias when defining time-varying exposures such as depression and antidepressant use and should consider using robust methods and sensitivity analyses to address these difficulties. Future studies with longer follow-up in diverse settings are needed to confirm our finding.

## Supporting information

**S1 Table. The number and percentage of included participants who met each of the criteria for depression in primary analysis.**
(DOCX)

**S2 Table. The number and percentage of participants who met each of the criteria for depression in the first sensitivity analysis.**
(DOCX)

## Author Contributions

**Conceptualization:** Fang Han, Ke-Gang Cao, Ying Gao, Xiao-Hua Zhou.

**Data curation:** Fang Han, Merilee A. Teylan, Lilah M. Besser.

**Formal analysis:** Tyler Bonnett.

**Methodology:** Fang Han, Willa D. Brenowitz, Lilah M. Besser, Yen-Chi Chen, Gary Chan, Ke-Gang Cao, Ying Gao.

**Project administration:** Fang Han, Merilee A. Teylan, Gary Chan, Ying Gao, Xiao-Hua Zhou.

**Software:** Tyler Bonnett.

**Supervision:** Fang Han, Willa D. Brenowitz, Merilee A. Teylan, Lilah M. Besser, Yen-Chi Chen, Gary Chan, Xiao-Hua Zhou.

**Validation:** Yen-Chi Chen.

**Writing – original draft:** Fang Han.

**Writing – review & editing:** Tyler Bonnett, Willa D. Brenowitz.

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
