## [Decision Letter · Decision Letter 0]

28 Oct 2019

PONE-D-19-27363

Associations between antidepressant use and incident mild cognitive impairment in older adults with depression

PLOS ONE

Dear Dr. Han,

Thank you for submitting your manuscript to PLOS ONE. After careful consideration, we feel that it has merit but does not fully meet PLOS ONE’s publication criteria as it currently stands. Therefore, we invite you to submit a revised version of the manuscript that addresses the points raised during the review process.

Two reviewers addressed a number of major and minor concerns about your manuscript. Please revise your manuscript carefully.

We would appreciate receiving your revised manuscript by Dec 12 2019 11:59PM. To enhance the reproducibility of your results, we recommend that if applicable you deposit your laboratory protocols in protocols.io, where a protocol can be assigned its own identifier (DOI) such that it can be cited independently in the future. For instructions see: http://journals.plos.org/plosone/s/submission-guidelines#loc-laboratory-protocols

We look forward to receiving your revised manuscript.

Kind regards,

Kenji Hashimoto, PhD

Academic Editor

PLOS ONE

Journal Requirements:

The NACC database is funded by NIA/NIH Grant U01 AG016976. NACC data are contributed by the NIA-funded ADCs: P30 AG019610 (PI Eric Reiman, MD), P30 AG013846 (PI Neil Kowall, MD), P50 AG008702 (PI Scott Small, MD), P50 AG025688 (PI Allan Levey, MD, PhD), P50 AG047266 (PI Todd Golde, MD, PhD), P30 AG010133 (PI Andrew Saykin, PsyD), P50 AG005146 (PI Marilyn Albert, PhD), P50 AG005134 (PI Bradley Hyman, MD, PhD), P50 AG016574 (PI Ronald Petersen, MD, PhD), P50 AG005138 (PI Mary Sano, PhD), P30 AG008051 (PI Thomas Wisniewski, MD), P30 AG013854 (PI M. Marsel Mesulam, MD), P30 AG008017 (PI Jeffrey Kaye, MD), P30 AG010161 (PI David Bennett, MD), P50 AG047366 (PI Victor Henderson, MD, MS), P30 AG010129 (PI Charles DeCarli, MD), P50 AG016573 (PI Frank LaFerla, PhD), P50 AG005131 (PI James Brewer, MD, PhD), P50 AG023501 (PI Bruce Miller, MD), P30 AG035982 (PI Russell Swerdlow, MD), P30 AG028383 (PI Linda Van Eldik, PhD), P30 AG053760 (PI Henry Paulson, MD, PhD), P30 AG010124 (PI John Trojanowski, MD, PhD), P50 AG005133 (PI Oscar Lopez, MD), P50 AG005142 (PI Helena Chui, MD), P30 AG012300 (PI Roger Rosenberg, MD), P30 AG049638 (PI Suzanne Craft, PhD), P50 AG005136 (PI Thomas Grabowski, MD), P50 AG033514 (PI Sanjay Asthana, MD, FRCP), P50 AG005681 (PI John Morris, MD), P50 AG047270 (PI Stephen Strittmatter, MD, PhD).

Reviewers' comments:

Reviewer's Responses to Questions

**Comments to the Author**

1. Is the manuscript technically sound, and do the data support the conclusions?

Reviewer #1: Yes

Reviewer #2: Yes

2. Has the statistical analysis been performed appropriately and rigorously? 

Reviewer #1: Yes

Reviewer #2: Yes

3. Have the authors made all data underlying the findings in their manuscript fully available?

Reviewer #1: Yes

Reviewer #2: Yes

4. Is the manuscript presented in an intelligible fashion and written in standard English?

Reviewer #1: Yes

Reviewer #2: Yes

5. Review Comments to the Author

Reviewer #1: The authors investigated whether the wide variety of approaches to characterizing antidepressant use, the differing criteria used to define depression, and the multitude of statistical procedures employed by previous studies may contribute to the inconsistent estimates of the association between antidepressant use and risk of Mild Cognitive Impairment (MCI) in older adults with depression. To investigate the association between antidepressant use and incident MCI, they implemented a time dependent Cox proportional hazard model and a fixed covariate model. Treating antidepressant use as a time-varying covariate, they found no significant association with incident MCI. In contrast, when antidepressant use was treated as a fixed covariate, they observed a significant association between having ever used antidepressants and lower risk of MCI. However, in the baseline-use only model, the association was non-significant. Taken together, they concluded that antidepressant use was not significantly associated with incident MCI among cognitively normal persons with depression.

This paper presents evidence that antidepressant use is not significantly associated with incident MCI among cognitively normal persons with depression. Furthermore, this paper demonstrates that methodological approaches play an important role in estimating the association between an exposure and results in epidemiological studies. The findings will be of interest to clinicians, as well as researchers in the field.

I have the following concerns.

1. Methods. P4, line 82. I think the Geriatric Depression Scale (GDS) used here is a short form 15 item GDS, not a 30-item GDS. It would be better to specify it at the first point.

2. Results. P7, line 154. “Participants had an average length of follow-up of five years and spent as many as 11 years in this study.”

The words, “spent as many as 11 years in this study” needs clarification.

3. Results. P8, Table 1. Schreiner et al. (2003) reported that a cut-off score of 6 for the GDS had a sensitivity of 0.97, a specificity of 0.96 for depression. On the other hand, the mean baseline GDS ranges from 3.0 and 4.1 for the subjects, indicating that most of the subjects were below the cut-off point of 6 for the GDS. In the study, subjects were considered depressed if they met at least two of the following five criteria at entry into the study: 1) self-reported active depression in the last two years, 2) depression or dysphoria symptoms as reported by a coparticipant on the NPI-Q, 3) GDS score of at least six, 4) clinically depressed mood based on clinician interview, or 5) a clinical diagnosis of active depression based on current UDS examination and the clinician’s best judgment. I think information on what percentage of subjects meet each criteria should be added to the results.

4. Results. P9, line 187-199. The authors performed two sensitivity analyses to address the impact of depression definition. The number of participants who met each depression definition should be added.

5. Discussion. P13, line 258. “A previous meta-analysis found that antidepressant users had increased odds of Alzheimer’s disease relative to non-users (OR=1.65).”

The authors should cite the relevant paper.

In conclusion, I enjoyed reading this paper. I am grateful that the editor and the authors have given me this kind of opportunity. I think the findings contribute to our understanding of the association between an exposure and an outcome.

Reviewer #2: I enjoyed reading this paper, which was thoughtful and clearly presented, and addressed an important methodological point in the literature. I have a few suggestions for consideration outlined below, divided by section.

Major Revisions

Introduction

Some readers might suggest that time-dependent models are the only acceptable way to analyse data whereby exposures can vary over time. I wonder if this question should be addressed in the Introduction? For instance, what is the purpose of doing a study that compares time-fixed models to a time-dependent model when we know we shouldn’t use time-fixed models? To address this issue, one could state that many studies DO use survival analyses that are sensitive to immortal time bias, and this study helps to demonstrate the potential issues with that approach. Alternatively, are there any downsides to using a time-dependent model in these types of studies? In general, it would be helpful to have a few more details as to the pros and cons of the various analytic techniques to set the stage for the rest of the paper.

Methods

Section 2.1: Consider stating here that many participants were seeking help for subjective cognitive impairment (if this is accurate). I believe that this feature of the study population may be important in interpreting results

In Section 2.3, the authors state that participants with dementia (who were never found to have MCI at UDS visits) were assumed to have passed through an MCI stage. I wasn’t clear whether these participants were included in the analyses?

Results

No suggestions, this section was clearly presented

Discussion

Overall, I find that there is a bit of conflation between the study’s aim of investigating the effects of different methodologies on determining the relationship between antidepressant use and incident MCI and the interpretation of results from a clinical perspective. For instance, much of the Discussion appropriately addresses the effects that various analytic techniques and measurement definitions may have on study outcomes, but in the final paragraph (discussing study strengths), the authors discuss how their primary results rely on a time-dependent proportional hazards model, along with other strengths of the study design. While these factors are true, they are not in keeping with the study aims. Overall, it would be helpful to clarify whether this study is primarily methodological or clinical in nature, and to be consistent throughout the manuscript.

The third and fourth paragraphs require some citations. Specifically, the first and second sentences of the third paragraph in the Discussion require citations regarding which studies use mixed model approaches and odds ratios. Also, in the fourth paragraph, the authors mention a meta-analysis that found an OR of 1.65 for antidepressant users developing Alzheimer’s. I don’t see a citation for this study.

The authors mention that many previous studies addressing similar research questions did not use survival analyses. However, there is at least one that used a comparable survival analysis: the Goveas et al. (2012) study, in which the authors used a Cox proportional hazards model to investigate the same question (though I believe time was fixed; they just examined baseline antidepressant use). This was a different population; specifically cognitively healthy older women who were NOT seeking help for cognitive complaints. It might be interesting to briefly compare the results from this study to the current study, as the methodological differences may explain the discrepancy.

Minor Revisions

Abstract: In the “Discussion” section of the abstract, the first sentence states “…and we suggested that antidepressant use being modeled as a time-varying covariate”. I believe this should state “…and we suggested that antidepressant use should be modeled as a time-varying covariate”.

Section 2.6: The third sentence is “And this time-dependent approach…”. I would suggest removing the “And” and starting the sentence with “This time dependent approach…”.

6. PLOS authors have the option to publish the peer review history of their article (what does this mean?). If published, this will include your full peer review and any attached files.

Reviewer #1: No

Reviewer #2: Yes: Kathleen S. Bingham

---

## [Author Response · Author response to Decision Letter 0]

14 Dec 2019

Dear editors and reviewers,

Thank you for your letter and the comments from the referees about our submitted research article titled "Associations between antidepressant use and incident mild cognitive impairment in older adults with depression," (PONE-D-19-27363). Your comments have been valuable to our revision process. We have fully accepted any minor comments on clarity of writing or formatting issues and changes to the text have been made accordingly. The largest revisions occurred in the Introduction and Discussion sections, where we have attempted to clarify the dual purposes of this work: 1) to perform a practical methodological comparison motivated by inconsistencies in previous estimates of this association and the persistence of time-fixed models in the literature even in the presence of time-varying covariates and 2) to highlight our own estimate of the association in question while commenting on the strengths of our modelling approach. Revised portions of the manuscript are marked in red.

Also, if possible at this stage, we want to change the title of our manuscript to a more appropriate one ——"Estimating associations between antidepressant use and incident mild cognitive impairment in older adults with depression".

Funding related text has been removed from the manuscript and is available in the Funding Statement section of the submission form.

Please find our responses to the reviewers’ comments below.

Reviewer 1: 

The authors investigated whether the wide variety of approaches to characterizing antidepressant use, the differing criteria used to define depression, and the multitude of statistical procedures employed by previous studies may contribute to the inconsistent estimates of the association between antidepressant use and risk of Mild Cognitive Impairment (MCI) in older adults with depression. To investigate the association between antidepressant use and incident MCI, they implemented a time dependent Cox proportional hazard model and a fixed covariate model. Treating antidepressant use as a time-varying covariate, they found no significant association with incident MCI. In contrast, when antidepressant use was treated as a fixed covariate, they observed a significant association between having ever used antidepressants and lower risk of MCI. However, in the baseline-use only model, the association was non-significant. Taken together, they concluded that antidepressant use was not significantly associated with incident MCI among cognitively normal persons with depression.

This paper presents evidence that antidepressant use is not significantly associated with incident MCI among cognitively normal persons with depression. Furthermore, this paper demonstrates that methodological approaches play an important role in estimating the association between an exposure and results in epidemiological studies. The findings will be of interest to clinicians, as well as researchers in the field.

Thank you for the positive feedback

I have the following concerns.

1. Methods. P4, line 82. I think the Geriatric Depression Scale (GDS) used here is a short form 15 item GDS, not a 30-item GDS. It would be better to specify it at the first point.

Response: The GDS used here is the Geriatric Depression Scale-15 (GDS-15). We now specify this in the manuscript. (P5, line 90)

2. Results. P7, line 154. “Participants had an average length of follow-up of five years and spent as many as 11 years in this study.” The words, “spent as many as 11 years in this study” needs clarification.

Response: We agree that our word choice was confusing. The maximum number of years of follow-up for any participant was 11 years. To avoid confusion, we have altered this sentence to simply report that average length of follow-up: “Participants had an average length of follow-up of five years in this study.” (P8, line 165)

3. Results. P8, Table 1. Schreiner et al. (2003) reported that a cut-off score of 6 for the GDS had a sensitivity of 0.97, a specificity of 0.96 for depression. On the other hand, the mean baseline GDS ranges from 3.0 and 4.1 for the subjects, indicating that most of the subjects were below the cut-off point of 6 for the GDS. In the study, subjects were considered depressed if they met at least two of the following five criteria at entry into the study: 1) self-reported active depression in the last two years, 2) depression or dysphoria symptoms as reported by a coparticipant on the NPI-Q, 3) GDS score of at least six, 4) clinically depressed mood based on clinician interview, or 5) a clinical diagnosis of active depression based on current UDS examination and the clinician’s best judgment. I think information on what percentage of subjects meet each criterion should be added to the results.

Response: We agree this is a useful addition. We have added a set of supplementary tables to the paper to address this issue and the issue below. A breakdown of the percentage of primary-analysis eligible subjects who met each of the criteria for depression is presented in Supplementary Table S1, referenced in the text in section 2.2 (P5, line 95-98). For your reference, we also present that table here. The table below displays the number and percentage (out of the full primary analysis population of 716) of subjects who met each of the criteria. As shown in the table, the number of participants (out of all those who met at least two criteria) who met the self-reported depression criterion was high (649/716; 90.6%). We do recognize that self-reported data may be less reliable, which was a motivation for requiring two depression criteria to be met as well as for our various sensitivity analyses where we address the impact of our composite definition of depression.

S1. The number and percentage of included participants who met each of the criteria for depression in primary analysis.

Criterion Total

(n=716)

Self-reported active depression in the last two years 649 (90.6%)

Clinical diagnosis of active depression based on current UDS examination and the clinician’s best judgement 498 (69.5%)

Depression or dysphoria symptoms as reported by a coparticipant on the NPI-Q 424 (59.2%)

GDS-15 score of at least 6 171 (23.9%)

Clinically depressed mood based on clinician interview 109 (15.2%)

4. Results. P9, line 187-199. The authors performed two sensitivity analyses to address the impact of depression definition. The number of participants who met each depression definition should be added.

Response: Thank you for the suggestion, this information has been presented in Supplementary Table S2, referenced in the text in section 3 (P10, line 197-200), which we also presented here. We felt that the supplementary tables were the best way to present this information because it allows the readers to quickly scan and compare the percentage who met each criterion in the primary analysis and the first sensitivity analysis. The second sensitivity analysis reduced the sample to only those who had a baseline GDS score of 6 or greater. The number of participants eligible for that sensitivity analysis was 231. We also now mention this in the text. Note that this is higher than the 171 participants reported to have met the GDS requirement as part of the primary analysis population, which indicates that there were 60 participants included in the primary analysis who met the GDS criterion but none of the other four criteria. 

S2. The number and percentage of participants who met each of the criteria for depression in the first sensitivity analysis.

Criterion Total

(n=307)

Self-reported active depression in the last two years 292 (95.1%)

Clinical diagnosis of active depression based on current UDS examination and the clinician’s best judgement 276 (89.9%)

Depression or dysphoria symptoms as reported by a coparticipant on the NPI-Q 247 (80.5%)

GDS-15 score of at least 6 124 (40.4%)

Clinically depressed mood based on clinician interview 94 (30.6%)

5. Discussion. P13, line 258. “A previous meta-analysis found that antidepressant users had increased odds of Alzheimer’s disease relative to non-users (OR=1.65).”

The authors should cite the relevant paper.

Response: We apologize for this oversite. This paper has now been cited. (P13, line 275)

In conclusion, I enjoyed reading this paper. I am grateful that the editor and the authors have given me this kind of opportunity. I think the findings contribute to our understanding of the association between an exposure and an outcome.

Response: Thank you very much for the helpful comments.

Reviewer 2: 

I enjoyed reading this paper, which was thoughtful and clearly presented, and addressed an important methodological point in the literature. I have a few suggestions for consideration outlined below, divided by section.

Major Revisions

Introduction

Some readers might suggest that time-dependent models are the only acceptable way to analyse data whereby exposures can vary over time. I wonder if this question should be addressed in the Introduction? For instance, what is the purpose of doing a study that compares time-fixed models to a time-dependent model when we know we shouldn’t use time-fixed models? To address this issue, one could state that many studies DO use survival analyses that are sensitive to immortal time bias, and this study helps to demonstrate the potential issues with that approach. Alternatively, are there any downsides to using a time-dependent model in these types of studies? In general, it would be helpful to have a few more details as to the pros and cons of the various analytic techniques to set the stage for the rest of the paper.

Response: Thank you for this very important comment. We agree with the suggestion that time-dependent models are the acceptable way of analyzing associations involving time-varying exposures. We have updated the introduction (P3-4，lines 64-70) to clarify our beliefs and highlight that our comparison is intended to show that the difference in hazard ratio estimates provided by time-fixed versus time-dependent models can be quite large in real-world examples. We have also added a citation to the second paragraph（P3，line 63）of the introduction as supporting evidence that the use of time-fixed models remains quite common even when exposures are known to vary over time. We do find that the persistence of these models which are susceptible to immortal time bias motivates a methodological comparison using real-world data.

Methods

Section 2.1: Consider stating here that many participants were seeking help for subjective cognitive impairment (if this is accurate). I believe that this feature of the study population may be important in interpreting results

Response: This is indeed accurate. Many subjects represented in the NACC database are either referred to an Alzheimer’s Disease Center or independently seek out an Alzheimer’s Disease Center due to concerns about their memory. We have added a sentence to section 2.1 to report this. (P4, line 77-79)

In Section 2.3, the authors state that participants with dementia (who were never found to have MCI at UDS visits) were assumed to have passed through an MCI stage. I wasn’t clear whether these participants were included in the analyses?

Response: These participants were included in the analyses. We have added a sentence to the end of section 2.3 to make this clearer (P5, line 112). We also reference this group in section 3: “There were 26 participants (20 ever-users and 6 never-users) who went directly from normal cognition to dementia without an intermediary MCI diagnoses, although for these participants an intermediate MCI stage was assumed to have occurred at the midpoint between their latest normal-cognition visit and their first diagnosis of dementia.” (P9, line 178-181)

Results

No suggestions, this section was clearly presented.

Response: Thank you.

Discussion

Overall, I find that there is a bit of conflation between the study’s aim of investigating the effects of different methodologies on determining the relationship between antidepressant use and incident MCI and the interpretation of results from a clinical perspective. For instance, much of the Discussion appropriately addresses the effects that various analytic techniques and measurement definitions may have on study outcomes, but in the final paragraph (discussing study strengths), the authors discuss how their primary results rely on a time-dependent proportional hazards model, along with other strengths of the study design. While these factors are true, they are not in keeping with the study aims. Overall, it would be helpful to clarify whether this study is primarily methodological or clinical in nature, and to be consistent throughout the manuscript.

Response: Thank you for this comment. As we have mentioned above, the bulk of our revisions have centered around this issue. We consider that this paper has a dual purpose, both methodological and clinical. We have added language in the title, abstract, introduction, and discussion sections to help clarify this point. We focus on methodologic comparisons as a potential explanation for discrepancy in findings on the association between anti-depressant use and cognitive impairment but also think it is valuable to comment on our findings in context of prior literature from a more clinical perspective. Like Reviewer 1 suggests, our aims are for this paper to be relevant to clinicians and researchers alike. We have made revisions throughout the discussion to be consistent with these goals.

The third and fourth paragraphs require some citations. Specifically, the first and second sentences of the third paragraph in the Discussion require citations regarding which studies use mixed model approaches and odds ratios. 

Response: Citations for both of these points have been added. (P13, line 262-264)

Also, in the fourth paragraph, the authors mention a meta-analysis that found an OR of 1.65 for antidepressant users developing Alzheimer’s. I don’t see a citation for this study.

Response: Our apologies for this oversite, which was helpfully noticed by both reviewers. We have added the appropriate citation. (P13, line 275）

The authors mention that many previous studies addressing similar research questions did not use survival analyses. However, there is at least one that used a comparable survival analysis: the Goveas et al. (2012) study, in which the authors used a Cox proportional hazards model to investigate the same question (though I believe time was fixed; they just examined baseline antidepressant use). This was a different population; specifically cognitively healthy older women who were NOT seeking help for cognitive complaints. It might be interesting to briefly compare the results from this study to the current study, as the methodological differences may explain the discrepancy. 

Response: Thank you for your careful reading. You are correct that Goveas et al. did use survival analysis in their research. We cite this work in the introduction as an example of a study which reported a much higher risk of cognitive impairment among depressed antidepressant users (HR=2.44 in the Goveas study for this group), which is a stark difference from our findings. We now make a brief comparison between our work and Goveas et al. in the third paragraph of the discussion and comment on some possible explanations for this difference. (P13, line 256-262）

Minor Revisions

Abstract: In the “Discussion” section of the abstract, the first sentence states “…and we suggested that antidepressant use being modeled as a time-varying covariate”. I believe this should state “…and we suggested that antidepressant use should be modeled as a time-varying covariate”.

Section 2.6: The third sentence is “And this time-dependent approach…”. I would suggest removing the “And” and starting the sentence with “This time dependent approach…”.

Response: We have revised these sentences. Moreover, we have carefully reviewed the manuscript for any additional instances where the word choice could be improved. If you have additional suggestions, we would be happy to receive them. 

Thank you very much for the helpful comments.

---

## [Decision Letter · Decision Letter 1]

3 Jan 2020

Estimating associations between antidepressant use and incident mild cognitive impairment in older adults with depression.

PONE-D-19-27363R1

Dear Dr. Han,

We are pleased to inform you that your manuscript has been judged scientifically suitable for publication and will be formally accepted for publication once it complies with all outstanding technical requirements.

With kind regards,

Kenji Hashimoto, PhD

Section Editor

PLOS ONE

Additional Editor Comments (optional):

Reviewers' comments:

Reviewer's Responses to Questions

**Comments to the Author**

1. If the authors have adequately addressed your comments raised in a previous round of review and you feel that this manuscript is now acceptable for publication, you may indicate that here to bypass the “Comments to the Author” section, enter your conflict of interest statement in the “Confidential to Editor” section, and submit your "Accept" recommendation.

Reviewer #1: All comments have been addressed

Reviewer #2: All comments have been addressed

2. Is the manuscript technically sound, and do the data support the conclusions?

Reviewer #1: Yes

Reviewer #2: Yes

3. Has the statistical analysis been performed appropriately and rigorously? 

Reviewer #1: Yes

Reviewer #2: Yes

4. Have the authors made all data underlying the findings in their manuscript fully available?

Reviewer #1: Yes

Reviewer #2: (No Response)

5. Is the manuscript presented in an intelligible fashion and written in standard English?

Reviewer #1: Yes

Reviewer #2: Yes

6. Review Comments to the Author

Reviewer #1: (No Response)

Reviewer #2: Thanks for the opportunity to review this paper again. The authors have carefully addressed my comments. I have two minor points. I do not need to see the manuscript again--these points are just for consideration.

Introduction, paragraph 3: A few grammatical suggestions:

"We have two goals for this study; the first is clinical and the second methodological: i) to evaluate the association between antidepressant use and cognitive impairment, and ii) to illuminate the benefits and pitfalls of different methodological approaches and apply relatively robust methods and definitions for inference"

If there is space, consider adding a brief sentence (and maybe an example) in the Intro explaining immortal time bias. Since this paper is also intended for a clinical audience it might be helpful to expand on this a bit more.

7. PLOS authors have the option to publish the peer review history of their article (what does this mean?). If published, this will include your full peer review and any attached files.

Reviewer #1: No

Reviewer #2: Yes: Kathleen S. Bingham

---

## [Editor Report · Acceptance letter]

10 Jan 2020

PONE-D-19-27363R1 

Estimating associations between antidepressant use and incident mild cognitive impairment in older adults with depression. 

Dear Dr. Han:

I am pleased to inform you that your manuscript has been deemed suitable for publication in PLOS ONE. Congratulations! Your manuscript is now with our production department. 

With kind regards,

on behalf of

Prof. Kenji Hashimoto 

Section Editor

PLOS ONE